# Line Tension and Drop Size Dependence of Contact Angle at the Nanoscale

**DOI:** 10.3390/nano12030369

**Published:** 2022-01-24

**Authors:** Waldemar Klauser, Fabian T. von Kleist-Retzow, Sergej Fatikow

**Affiliations:** Division Microrobotics and Control Engineering, Department of Computing Science, University of Oldenburg, D-26129 Oldenburg, Germany; fabian.von.kleist-retzow@uni-oldenburg.de (F.T.v.K.-R.); sergej.fatikow@uni-oldenburg.de (S.F.)

**Keywords:** adhesion, liquid metal, Galinstan, contact angle, scanning electron microscope, modified Young’s equation, line tension

## Abstract

Despite considerable research efforts, the influence of contact line tension during wetting at the nanoscale and its experimental determination remain challenging tasks. So far, molecular dynamics simulations and atomic force microscope measurements have contributed to the understanding of these phenomena. However, a direct measurement of the size dependence of the contact angle and the magnitude of the apparent line tension has not been realized so far. Here, we show that the contact angle is indeed dependent on the drop size for small drop diameters and determine the magnitude of the apparent line tension via liquid-metal based measurements of advancing and receding contact angle inside a scanning electron microscope. For this purpose, a robotic setup inside an electron microscope chamber and oxide-free Galinstan droplets—produced via an electromigration-based and focused ion beam irradiation-assisted process—are employed. Using the first-order correction of Young’s equation, we find an apparent line tension value of 4.02 × 10^−7^ J/m for Galinstan© on stainless steel.

## 1. Introduction

Understanding wetting phenomena at the micro- and nanoscale is vital for a broad range of applications, including digital microfluidics [1,2], additive manufacturing [3], coating technologies [4], and tribological systems [5]. In contrast to the macroscale, where the liquid–solid contact is governed by the three surface tensions of the solid, the liquid, and the gas phases, at the nanoscale one also has to take into account the line tension, i.e., the force of the tensile or compressive strength of the one-dimensional three-phase contact line [6]. In analogy to surface tension, which seeks to reduce the area of the interfaces, the line tension seeks to reduce the free energy of the solid–liquid–gas system by adjusting the length of the contact line of the liquid drop.

Line tension was thermodynamically described by Gibbs as the excess free energy per unit length of the contact line [7]. Its cause is attributed to the imbalance of intermolecular forces in the three phase contact region [6]. It leads to a rebalance of interfacial forces in this region, and thus to a modified form of Young’s equation [8]:
(1)$$\cos\theta =\frac{\gamma_{\mathrm{SG}}-\gamma_{\mathrm{LS}}}{\gamma_{\mathrm{LG}}}-\frac{\tau }{{r\gamma }_{\mathrm{LG}}}{=\cos\theta }_{\infty }-\frac{\tau }{{r\gamma }_{\mathrm{LG}}}$$
where *γ*_LG_, *γ*_SG_, and *γ*_LS_ are the liquid–gas, solid–gas, and liquid–solid interfacial tensions and *θ*_∞_ and *θ* are the contact angles of a macroscopic droplet and a droplet with the contact radius *r*, respectively. Here, *τ* represents not only the thermodynamic line tension, but also the curvature-dependent surface tension and line contribution effects [9,10,11,12], and thus, it is referred to as the apparent line tension.

From Equation (1) it can be seen that the contact angle becomes size dependent at the nanoscale. Furthermore, one can determine the apparent line tension by measuring the contact angle of the drops with different diameters at this scale.

However, the reduction of the droplet size is associated with some difficulties. For example, the influence of the evaporation rate increases strongly as the droplet diameter decreases. The possibilities for controlled manipulation of the droplets become severely limited and determining the contour angle requires high-resolution microscopes. Thus, despite many efforts for several decades, the experimental determination of the apparent line tension has remained a difficult task. Positive as well as negative values have been reported, spanning over several orders of magnitude from 10^−12^ to 10^−5^ J/m [6,13,14,15,16,17,18,19,20,21]. So far, most methods rely on AFM measurement of the drop shape and molecular dynamics (MD) simulations. Recently, Zhao et al. have determined the apparent line tension by acquiring the topography of nanoscale 1-butyl-3-methylmidazolium iodide drops via atomic force microscope (AFM) measurements and the subsequent three-dimensional cap fitting [22]. While the results show *τ* values that are in agreement with the literature, this technique (as well as others based on AFM measurements) suffers from not being able to accurately capture the behavior in the liquid–solid–gas contact point due to the finite size of the AFM cantilever tip and having to rely on cap fitting and extrapolation of the contact angle and radius for data analysis, which might not be accurate—especially for contact angles > 90°.

To address these issues, we have previously presented the use of a liquid metal as a liquid with a barely measurable evaporation rate [22] and demonstrated robotically assisted contact angle measurements in a scanning electron microscope (SEM) [23]. To address the issue of oxide layer growth on the liquid metal droplets, which would prevent them from being useful for contact angle measurements [24], our experiments take place in the low-oxygen vacuum atmosphere of an SEM chamber.

Thus, in this work, we are able to investigate the dependence of the contact angle on drop size via direct measurement of the advancing and receding contact angle of liquid metal droplets with different diameters inside an SEM. From those measurements, we determine the magnitude of the apparent line tension using Equation (1). Our results show that the contact angle is (a) indeed size dependent once the contact area reaches the nanoscale, and (b) the magnitude of the line tension of Galinstan is in the range of 10^−7^ J/m.

## 2. Materials and Methods

The experiments are carried out inside a high-resolution scanning electron/focused ion beam (FIB) microscope TESCAN Lyra FEG (TESCAN, Brno, Czech Republic). A robotic nanomanipulation setup inside the vacuum chamber is used for the manipulation of the liquid metal droplets. The characteristics of the robotic setup are described in more detail in our previous work [25]. In the following, we use the term “droplet” where the liquid properties of the liquid metal are in focus, whereas we resort to the term “sphere” where the geometry is the main point.

The liquid metal droplets are produced via an electromigration process [26] during which drops of an alloy with a mass ratio of 68.5% Ga, 21.5% In, and 10% Sn (equivalent to Galinstan©) are grown at the apex of an electrochemically etched and FIB-customized tungsten tip (end effector), which is electrically connected to a source measurement unit (SMU, Keysight B2901A) and mechanically mounted to the nanomanipulation setup. For growth of the liquid metal droplet, an electrostatic potential between the substrate and the tip is applied by the SMU. When an electrical current is flowing, the liquid metal mass flow occurs in the direction of the cathode, while the tip is in contact with the liquid metal reservoir. By using the SMU, the electrical current is maintained at a constant to keep the liquid metal mass flow constant as well. When the desired diameter of the sphere is reached, the electrical source is switched off and the tip is removed from the liquid metal reservoir (see Appendix A). Since the non-conductive surface oxide does not participate in the electromigration process, the produced droplet consists of oxide-free liquid metal, which remains stable within the prevailing vacuum environment of the SEM.

For fixation of the resulting droplet at a designated location on the tungsten tip, a newly developed technique of forced wetting by ion implantation is utilized [27]. Since it is important for the contact angle measurements that only the droplet touches the surface to be measured, the positioning of the droplet at the tip is crucial. By using the technique, a specific location on the tip can be wetted with the liquid metal. This spot then serves as the preferred point at which the sphere adheres to the tip. Another requirement for the design of the manipulator is that, after the completion of a series of measurements, the size of the sphere can be reduced by using FIB milling and thus the radius of the sphere can be exactly controlled. For this, the sphere must be visible for both the SEM beam and the FIB beam. The chosen design allows this by placing the sphere at the front and side of the tip (Appendix A). When reducing the size of the sphere by FIB milling, additional care must be taken to ensure that the irradiated area does not contain any part of the tip, otherwise further wetting may occur and the sphere may be destroyed. In this case, destruction means that the liquid metal no longer has a spherical shape but, for example, adheres to the tip in an elongated form (Appendix A). This makes the droplet unusable for further contact angle measurements.

The end effector created in this way and consisting of an oxide-free liquid metal sphere attached to a tungsten tip is used for all subsequent experiments.

The stainless steel sample (AISI420, Goodfellow, UK) is cleaned with isopropyl alcohol in an ultrasonic bath, followed by an oxygen plasma treatment to remove surface contaminations.

The method of measuring the contact angles was repeated following the same procedure for every series. This study followed the scientific procedure of the height variation method [28]. First, the manipulator was prepared as described above. The manipulator (see Figure 1III) was then brought into contact with the stainless steel surface several times, each time measuring the advancing (see Figure 1I) and receding angles (see Figure 1II). Four measurement series are shown in Figure 1, depicting an advancing angle (I), a receding angle (II), and the manipulator (III) for each series, respectively. The different contact angles were determined by a linear fit of two straight lines crossing at the three-phase point. The straight lines were tangential to the two two-phase boundaries (liquid–solid and liquid–gas). Further information on the measurement method can be found in [23]. A more detailed analysis of the measurement results obtained is given in the next chapter.

During the measurements, several things should be considered. For example, the influence of the SEM beam can lead to a change in adhesion. Thus, especially when measuring with smaller spheres, care must be taken to use the same magnification as when measuring larger spheres. Due to the increased irradiation dose at higher magnification, electron beam-induced depositions (EBiD) are more likely to be formed [29] and can lead to a significantly stronger adhesion of the liquid metal to the surface. Thus, the increased accuracy from measuring the contact angle at a higher magnification must be weighed against the stronger influence of the electron beam. Consequently, these measurements must be performed faster or at a lower magnification relative to the sphere radius, which in turn leads to inaccuracies.

In addition, the increasing influence of the oxide layer with longer measuring times must also be taken into account. Therefore, the measurements were performed within a maximum of ten minutes after the preparation of the sphere to keep this influence low. More details about the contact angle measurement technique and the influence of the oxide layer can be found in previous work [23].

## 3. Results

Measurements of advancing and receding contact angles for different droplet sizes on a stainless steel substrate have been performed by the height variation method [28], as described above. Knowing the advancing and receding contact angles, the equilibrium angle *θ_C_* can be calculated according to following equations [30]:(2)$$\theta_{c}=arccos\left(\frac{r_{A}\;cos\theta_{A}+r_{R}\;cos\theta_{R}}{r_{A}+r_{R}}\right)$$
(3)$$r_{A}={\left(\frac{sin^{3}\theta_{A}}{2-3\;cos\theta_{A}+cos^{3}\theta_{A}}\right)}^{1/3}$$
(4)$$r_{R}={\left(\frac{sin^{3}\theta_{R}}{2-3\;cos\theta_{R}+cos^{3}\theta_{R}}\right)}^{1/3}.$$

Contact angle hysteresis is caused by inhomogeneities of the surface, and thus Equations (2)–(4) can be derived by incorporating the contribution of the line tension into Young’s equation and then assuming that the resistance to motion for an advancing drop is equal to the resistance of the motion in a receding drop because both of these resistances are a result of the pinning of the contact line to similar surface irregularities [30].

Figure 2 shows the behavior of the advancing, receding, and equilibrium contact angles with decreasing liquid metal droplet sizes. The error bars represent the standard deviation of the measurement results around the mean value. While the advancing angle is hardly influenced by the drop size, it can be seen, despite the variation of individual values, that, overall, the receding angle drops with the decreasing size of the liquid metal sphere—starting from a critical diameter of around 10–12 µm—and thus, the equilibrium contact angle shows a similar behavior. For example, the difference in the equilibrium contact angle can amount to more than 50° between droplets of around a 12 µm diameter and ones with a diameter of 2 µm. From these findings, it can be expected that the receding and equilibrium contact angles decrease even more for smaller droplets.

To quantify the magnitude of the apparent line tension *τ*, we plot the cosine of *θ_C_* versus the diameter of the liquid metal droplets and fit Equation (1) to the data using the contact angle of a droplet with a 25 µm diameter as *θ*_∞_ (see Figure 3a). The surface tension of Galinstan© *γ_LG_* has been assumed to be 0.535 J/m^2^ [28]. From this fit, we determined a value for the apparent line tension, *τ* = 4.02 × 10^−7^ J/m. Furthermore, for better display of the validity of the measurement results, *θ_C_* versus one, divided by the radius of the spheres, was plotted too (see Figure 3b). The solid red line corresponds to the previously determined *τ*. The additional two dotted lines correspond to 2 *τ* and 1/2 *τ*. Thus, more than 90% of the data points are located in the funnel that spans between 2.01 × 10^−7^ J/m < *τ* < 8.04 × 10^−7^ J/m. The apparent line tension is also influenced by the surface topography, on top of the other properties of the liquid, solid, and gaseous phases. In our previous work we extracted an S_a_ value of 149.5 nm from AFM topography measurements on the surface of a similar stainless steel sample [23]. This relatively high surface roughness value might be responsible for the increased scattering of cos(θ) with the decreasing drop size in Figure 3b, according to the findings of Lin et al. [31]. Furthermore, the dependence of the line tension on Tolman length and the position of the liquid–solid dividing interface has been shown via MD simulations by Schimmele et al. [10] and Zhang et al. [11]. Due to the experimental nature of the presented approach, it is not exactly possible to determine the position of the liquid–solid dividing interface during the measurements, which may have also contributed to the comparable large scattering of cos(θ) seen in Figure 3b. Although the value of the apparent line tension found here is relatively high compared to the results for liquids mainly governed by van der Waals interactions, it seems reasonable for an ionic liquid such as Galinstan© with much stronger interactions and a much higher surface tension than most other liquids, and is well inside the range of values found reported in the literature (10^−12^ to 10^−5^ J/m [6,13,14,15,16,17,18,19,20,21]).

## 4. Discussion

Due to its low magnitude and it being significant only for very small droplets, the experimental investigation of line tension has remained a challenging task. It was even considered a myth, at least for macro and microscopic droplets [33,34]. However, when moving down to the nanoscale, its contribution to the wetting behavior increases to a point where it can influence the wetting behavior. This fact is vital for the future development and application of devices based on nanomaterials. Our measurements present a method to experimentally assess the behavior of the contact angle and the magnitude of the apparent line tension for drops of liquid metals with diameters well below 10 µm, and even for sub-micrometer droplets in future experiments. Although the results cannot be directly transferred to other liquids, the general behavior can be expected to be similar as Galinstan©, given that it behaves as a Newtonian liquid when it is not covered by an oxide layer. Furthermore, for the first time, to the best of our knowledge, we found an apparent line tension value for Galinstan© of *τ* = 4.02 × 10^−7^ J/m, which other approaches and future experiments can build on.

Our findings in Figure 2 indicate that the receding and equilibrium contact angles would decrease even more for droplets with diameters below 1 µm. Our future work will focus on the investigation of droplets on this size scale, as well as on the single and few asperity contact mechanics of micro and nanoscopic liquid metal droplets.

## Figures and Tables

**Figure 1 nanomaterials-12-00369-f001:**
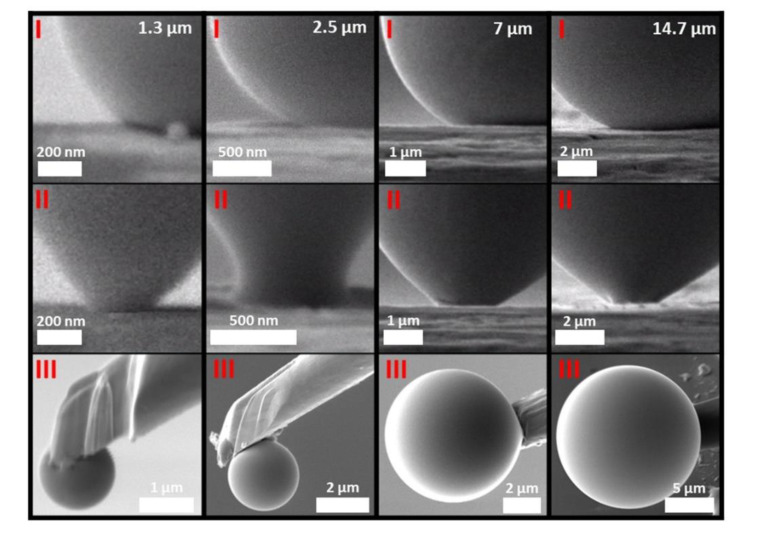
SEM micrographs presenting four different contact angle measurements series (columns from left to right) with liquid metal droplet diameters of 1.3 µm, 2.5 µm, 7 µm, and 14.7 µm, respectively; the upper row (**I**) depicts the advancing contact angles of each measurement, the middle row (**II**) depicts the receding contact angles, while the last row (**III**) the respective manipulator is illustrated.

**Figure 2 nanomaterials-12-00369-f002:**
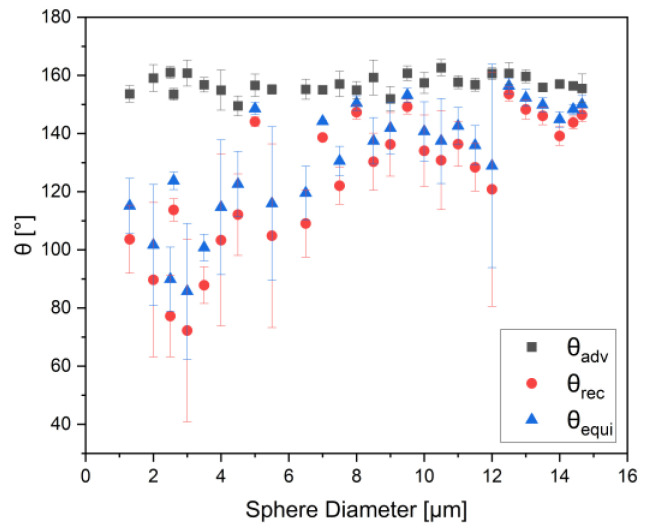
Values of advancing, receding, and the equilibrium contact angle calculated according to Equations (2)–(4) for liquid metal spheres with different diameters; the trend shows decreasing receding and equilibrium contact angle with decreasing sphere size, whereas the advancing contact angle doesn’t show a significant change with decreasing drop size.

**Figure 3 nanomaterials-12-00369-f003:**
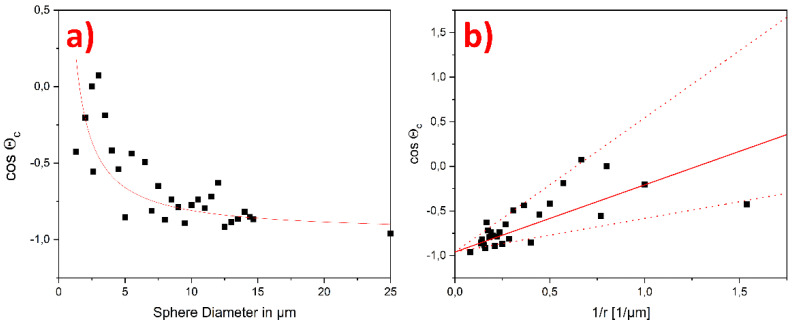
Cosine of the contact angle plotted vs. the sphere size, (**a**) the red line is a fit of Equation (1) from which *τ* = 4.02 × 10^−7^ J/m is determined, the surface tension of Galinstan© *γ*_LG_ has been assumed to be 0.535 J/m^2^ from reference [32]. (**b**) The same data points are fitted over one divided by the radius of the spheres. To better determine the validity of the measurement results, two additional dotted lines corresponding to 2*τ* and 1/2 *τ* were added.

## Data Availability

Experimental data is available from the authors upon reasonable request.

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
