# Peer review of "Line Tension and Drop Size Dependence of Contact Angle at the Nanoscale"

_nanomaterials, 2022, doi:10.3390/nano12030369_

Round 1
Reviewer 1 Report
In the manuscript, the authors describe their experiment on measuring the contact angle of Galinstan (a metallic alloy) droplets on stainless steel. They use a scanning electron microscope with a robotic setup.
From the measured contact angle variation on the droplet size, they extract the apparent line tension. The manuscript is concise and well written.
I believe the manuscript can be suitable for Nanomaterials, but prior to that, I suggest that the authors amend a few details and better place their findings into a broader context of line tension of metals.
Query 1:
The apparent line tension generally depends on the surface topography, such as the roughness. Can the surface roughness be estimated in the experiment and commented on whether it plays a role in the measured line tension?
Query 2:
Line tension depends not only on the liquid but also on the substrate, which should be emphasized more in the discussion section. The measured line tension is certainly much larger than what is typically reported for aqueous or organic liquids. Is such a large line tension value expected for metals? I suggest that the authors carefully check the literature for reports of the line tension for other metals.
Query 3:
Figure 2: What hits the reader in the eye is that the blue and red fits do not really look good - there are clearly more data points on one side of each curve than the other. Related to this, why is an exponential function used for the fitting rather than eq (1) in the introduction?
Query 4:
I believe that Fig 3. would be much more revealing if plotted as cos (theta_c) versus 1/diameter, which would enable a linear fit of Eq. (1) and, by that, a much better demonstration of its validity.
Query 5:
Could a simple one-sentence description be given for Equations 1-3, which should, in fact, be labeled as 2-4? I think that would be useful for the reader.
Query 6:
typos - line 34: "It's", line 76: 10-7 J/m
Reviewer 2 Report
The present work employed scanning electron microscope to measure the contact angle of droplets and determine the apparent line tension defined in the modified Young’s equation. Overall, it is an interesting work. However, there are a couple of critical issues need to be addressed.
- The accuracy of the measurements of advancing and receding contact angles is very crucial for the determination of line tension. However, the authors only say that they are performed by the height variation method. More descriptions are needed. Is circular fit or linear fit of the liquid-gas interface used to determine contact angle?
- The magnitude of the apparent line tension determined in this work is much larger than most of results reported in the literature. More discussions and explanations are required. As reported in references [10,11], the apparent line tension is highly dependent on the position of the liquid-solid dividing interface. Discussion on this point is also needed.
Author Response
The present work employed scanning electron microscope to measure the contact angle of droplets and determine the apparent line tension defined in the modified Young’s equation. Overall, it is an interesting work. However, there are a couple of critical issues need to be addressed.
1. The accuracy of the measurements of advancing and receding contact angles is very crucial for the determination of line tension. However, the authors only say that they are performed by the height variation method. More descriptions are needed. Is circular fit or linear fit of the liquid-gas interface used to determine contact angle?
Thank you very much for this comment. Because we used this procedure for a long time, we forgot to explain it enough. We used linear fitting of the phase boundaries. Circular fit is hard to apply because the contact angles are sometimes smaller than 90° and taking a picture of the whole sphere would take too much measurement time and could have an impact on the results (because of more depositions through the electron beam).
Added the following to the manuscript:
“The different contact angles were determined by a linear fit of two straight lines crossing at the three-phase point. The straight lines were tangential to the two two-phase boundaries (liquid-solid and liquid-gas). Further information on the measurement method can be found in [23]. „
2. The magnitude of the apparent line tension determined in this work is much larger than most of results reported in the literature. More discussions and explanations are required. As reported in references [10,11], the apparent line tension is highly dependent on the position of the liquid-solid dividing interface. Discussion on this point is also needed.
As Galinstan is an ionic liquid with much stronger forces acting inside the drop than in organic liquids, and with its surface tension being 0.535 J/m2 [28] (the surface tension of water is around 0.072 J/m2 at 25 °C) a much higher line tension value is reasonable to be expected.
Following text has been added to the manuscript (line 202 – 207):
Furthermore, the dependence of line tension on Tolman length and the position of the liquid-solid dividing interface has been shown via MD simulations by Schimmele et al. [10] and Zhang et al. [11]. Due to the experimental nature of the presented approach it is not possible exactly determine the position of the liquid-solid dividing interface during the measurements which may have also contributed to the comparable large scattering of cos(θ) seen in Figure 3 b).

Round 2
Reviewer 2 Report
I agree its publication in the present form.